# Psychedelic-Assisted Therapy in Palliative Care—Insights from an International Workshop

**DOI:** 10.3390/healthcare13182275

**Published:** 2025-09-12

**Authors:** Anna Schuldt, Ian C. Clark, Yasmin Schmid, Michael Ljuslin, Christopher Boehlke, Sivan Schipper, Megan B. Sands, David Blum

**Affiliations:** 1Center for Palliative Care, University Hospital Zurich, 8091 Zurich, Switzerland; ian.clark@usz.ch (I.C.C.); david.blum@usz.ch (D.B.); 2University of Zurich, 8006 Zurich, Switzerland; 3Clinical Pharmacology and Toxicology, University Hospital Basel, 4031 Basel, Switzerland; yasmin.schmid@usb.ch; 4Department of Clinical Research, University Hospital Basel and University of Basel, 4031 Basel, Switzerland; christopher.boehlke@unibas.ch; 5Department of Pharmaceutical Sciences, University of Basel, 4056 Basel, Switzerland; 6Palliative Medicine Division, Department of Rehabilitation and Geriatrics, Geneva University Hospitals, 1226 Geneva, Switzerland; michael.ljuslin@hug.ch; 7University of Geneva, 1205 Geneva, Switzerland; 8Palliative Care Center Basel, Bethesda Hospital, 4052 Basel, Switzerland; 9Center for Palliative Care, Spital Uster, 8610 Uster, Switzerland; sivan.schipper@spitaluster.ch; 10Department of Palliative Medicine, Calvary Mater Newcastle, Waratah 2298, Australia; 11Radwick Clinical School, Clinical Medicine, University of New South Wales, Sydney 2052, Australia

**Keywords:** palliative care, life-limiting illness, psychedelic, psychedelic-assisted therapy, mental health

## Abstract

**Background:** Evidence is growing that psychedelic substances have positive effects in the setting of Palliative Care (PC), focusing on special needs in this patient population. After a scoping review of the literature, no published expert recommendations guiding best practice for psychedelic-assisted therapy (PAT) towards the end of life were identified. **Objective:** To draw conclusions from first-hand experienced professionals on PAT in PC (PATPC). **Setting, Design, Participants:** An international workshop with experts was held in Wasserfallen, Switzerland. A thematic analysis of a semi-structured, questionnaire-based qualitative study with 13 experts in PC, oncology, psychiatry/psychology, and PAT from Europe, the United States, and Oceania was made. **Measurements:** The questionnaire was designed to elicit the participant’s perspectives on (A) special considerations on PATPC, (B) specific characteristics of PATPC (versus mental illness), and (C) the relevance of these differences during preparation, substance dosing session, and integration in PATPC. **Results:** (A) Special Considerations included (non-medicalized) setting, potential need, and possibility to reduce preparation time. (B) Distinguishing characteristics included the patient’s intrinsic motivation for treatment success, the importance of anxiety, depression, and spiritual distress as indications for PATPC, and the importance of sufficient integration of the psychedelic experience into life in the face of limited time due to the life-limiting illness. (C) Flexibility in setting and timing of preparation, choosing the appropriate dosage of the psychedelic substance depending on the patient’s intended focus, low/medium (relational issues), higher for transcendental experiences, considering mental capacity and vulnerability for the individual. In addition, respondents noted that for therapists, knowledge about transcendental states, such as mystical experiences, existential aspects of life-threatening illness, and the role of therapists’ own self-experience/inner work, as well as good knowledge of the theoretical basis for treatment, was highlighted. **Conclusions:** This study highlights special considerations for PAT PC and could be a first step towards specific treatment recommendations (guidelines) for PATPC.

## 1. Introduction

Life-threatening diseases affect an individual on many levels of being. Besides physical suffering, there is often a striking negative impact on emotional, psycho-social, and spiritual well-being, manifesting as existential distress and suffering. Patients may present with demoralization syndrome, adjustment disorder, depression, anxious mood, fear of death, and pain on both physical and psychological levels [1,2,3,4,5]. Major depression affects up to 45% of patients [5] and demoralization has a prevalence of 13% to 18%, and both are associated with an increase in overall symptom burden, as well as a wish for hastened death [1,6]. CareSearch [7] defines suffering in palliative care (PC) as


*a complex, multidimensional experience encompassing existential suffering, spiritual distress, hopelessness, and depression, requiring an inclusive, patient-centred approach that addresses both physical and non-physical aspects of well-being.*


Suffering experienced by people towards the end of life (last months of life) can cause a decrease in quality of life and may decrease therapeutic adherence [1,2,3,4,5,6,8]. Our focus in terms of context for psychedelic-assisted therapy (PAT) in PC (PATPC) was provided by participants, which includes formal psychiatric conditions as well as suffering and existential distress. We are not offering a set of defined indications for PATPC, but rather reporting on the survey respondents’ views on PATPC.

Palliative Care provides multidisciplinary, holistic, person-centered care, focused on symptom management and quality of life. As such, it may often provide effective and sufficient options for treating physical and emotional/existential suffering [8]. It must also be noted that some people do not report emotional and existential suffering beyond their own coping resources. Although evidence-based interventions, such as supportive therapies including dignity therapy and meaning-making therapy, are available [9,10,11]. Effective and lasting treatment approaches for addressing the psycho-spiritual dimension of suffering are still, for some, partly missing. Pharmacologic interventions for depression usually comprise antidepressants [12,13,14]. Benzodiazepines are at times used for anxiety, with little evidence of benefit [15]. Both of these drug modalities address the symptoms but not the cause of suffering, along with a relevant burden of side effects [14,15,16,17,18].

Psychedelic-assisted therapy (PAT) had its first zenith in clinical use and clinical research during the 1950s up through the early 1970s. Earlier studies from the 1960s and 1970s using psychedelics (mainly lysergic acid diethylamide (LSD)) showed promising results, on both psychological well-being as well as on physical pain [19,20,21,22] in people with life-limiting cancer. After no research in humans for decades, psychedelic research is currently experiencing a revival. Several recent pilot studies showed that classic serotonergic psychedelics like psilocybin and LSD could lead to a rapid, significant, and sustained reduction in anxiety, depressive symptoms, and existential distress in patients suffering from life-threatening illnesses (mainly cancer) and thus foster an increase in quality of life and meaning at the end of life [23,24,25,26,27,28,29,30,31,32].

Despite the growing evidence on the effectiveness of psychedelic substances in PC, after a scoping review of the literature, we found no published expert PAT treatment guidelines or recommendations available focusing on the unique needs of patients embarking on PATPC. A recent Cochrane review [33] concluded that evidence from intervention trials suggests a reduction of anxiety and depression with low certainty of evidence, i.e., there was a positive signal but in low numbers of patients. Further trials are ongoing.

Clinical trials included in that Cochrane review [33] were not limited to people with life-threatening illness towards the end of life, nor did it focus on recommendations on setting, timing, indications, or the relevance of therapist knowledge of spiritual pain for PATPC. Other sources of evidence underline the need for detailed therapeutic protocols for PATPC [34], evaluation of the safety of psychedelic interventions generally [35] and treatment risk-benefit balance [36]. We hypothesized that the unique circumstances (both physical and existential) in which people with life-threatening illness find themselves towards the end of life may require an adapted approach. There seems to be a therapeutic imperative to understand the potential of psychedelic medicines, particularly when physician-assisted dying is one current option for patients in this situation in many jurisdictions [37].

With the aim of clarifying the attitudes and phenomenology of clinicians experienced with PAT for patients towards the end of life, we conducted this thematic study with experienced professionals to elaborate on the following areas: (A) Are there special considerations on PAT in the PC setting that should be applied? (B) Which specific characteristics distinguish PAT in palliative patients from PAT in patients with mental illnesses without a life-threatening disease? (C) To what extent are these differences relevant during the three stages of PAT (preparation, substance session, and integration)?

## 2. Methods

### 2.1. Study Design and Context of the Study

An international training course for PATPC, which took place in Wasserfallen, Switzerland, from 5 to 7 April 2024. The training course encompassed lectures by experienced PAT therapists with knowledge of PC, lectures by researchers involved in psychedelic studies, as well as workshops for the participants. This gathering of clinicians and researchers with combined experience of many decades presented a valuable opportunity to gain insight into clinically embedded experience. We conducted a semi-structured written questionnaire-based qualitative study with experts in the field of PAT and PC, following the CROSS guidelines [38]. Different opinions were reconciled in discussion, and direct feedback to participants was not planned, but the publication will be shared.

### 2.2. Ethical Considerations

We used a questionnaire to capture non-health-related data (i.e., opinions on PATPC) from participants, and thus our study did not meet the criteria of research involving human beings as defined by Swiss law (Human Research Act, Art. 2; Ordinance on Human Research with the Exception of Clinical Trials, Art. 6) and for this reason our study was not subject to formal ethical review. Nevertheless, we take ethics very seriously and conducted all study procedures accordingly. We obtained written consent from all the participants prior to participation, and the questionnaire was anonymized.

### 2.3. Participants

Workshop participants had expertise in PC, oncology, and psychiatry, with some participants being trained in PAT in either clinical and/or research settings. Due to the high number of applications for the course, participants were selected according to the following criteria: experience (i.e., number of years practicing PAT), motivation (i.e., motivational statement on application reviewed by workshop hosts), and profession (i.e., MD, psychiatrist, or psychologist).

### 2.4. Conceptual Design of the Questionnaire

The initial draft of the written questionnaire was based on the therapeutic procedures described in clinical trials [33] and was written by A.S. (psychiatrist, study assistant), subsequently refined by S.S. (PC physician working with PAT, researcher) and D.B. (PC physician, researcher), as well as by other members of the research team with a background in PC and psychology. The questionnaire was refined, and different opinions were solved by consensus. We did not pre-test for clarity or bias specifically, but tested the questionnaire in our group. Theoretical saturation was not sought due to the limited possibilities to gain more data. The questionnaire included three main sections, asking for (1) professional background, (2) general considerations of PATPC, and (3) specific aspects of PATPC (see Appendix A).

### 2.5. Data Collection

The study questionnaire was distributed to all presenters and participants (N = 40) during the workshop via paper packets. Answers were transcribed and sorted according to the respondents’ level of experience.

### 2.6. Data Analysis

The responses to the open-ended questions of the questionnaire were analyzed using the framework method for the analysis of qualitative data in multi-disciplinary health research [39]. Differences according to professional background and experience with PAT were compared. Thematic analysis was performed by A.S. No independent coding or software was used.

## 3. Results

### 3.1. Study Sample

Of the 40 workshop participants, 13 (30%) returned the questionnaire, of which nine had a medical degree (two psychiatrists, three oncologists, two PC specialists, and two had another specialization), and four were trained in psychology. Five participants had extensive experience in PAT with >5 years to >20 years of experience. Those were considered experts in PAT. The other participants had <5 years or no experience with PAT (Table 1). Comparing the answers of experienced vs. non-experienced participants and therapists with somatic background vs. psychological/psychiatric background showed no relevant differences. Data are presented in Table A1 (see Appendix A). Results are grouped under three main themes presented in the methodology.

(A)Special Considerations on PAT in the PC Setting

### 3.2. Differences Between the Setting of PATPC vs. Non-PC PAT

Five out of thirteen (39%) participants reported the patient’s health condition as having a crucial impact on the set and setting. Also, six of thirteen (46%) mentioned the amount of time and, thus, the length of therapy available in each of the settings, with less time in PC due to the life-limiting illness. As a further aspect, the spiritual/existential dimension of the psychedelic experience was named in 4/13 answers (31%), emerging spontaneously with a higher frequency in patients with life-limiting diseases.

Two of the participants stated additionally that in the PC setting, people tend to be psychologically more stable compared to psychiatric patients. Also, it was mentioned twice that in PC, it might be more relevant to involve family and relatives.

Quote (#3 MD, Psychiatrist, >20 y PAT exp.): “[The main differences between the settings of PATPC and non-PATPC are the] time for treatment, the clear focus on a specific issue, the existential dimension, the non-influence of the most important topic, i.e., the fatal illness”.

### 3.3. Parameters Based on Which a Suggestion for PATPC Could Be Considered

It was considered to be crucial that a patient shows intrinsic motivation for PAT (7/13, 54%), especially in association with anxiety and fear of death (6/13, 46%). Besides the presence of fear of death, anxiety, depression, and existential distress, the suffering due to unsolved biographical issues was considered a possible target for PAT as well.

Quote (#2 Psychologist, >20 y PAT exp.): “[If there is a] yearning for ‘living well’ as death becomes more imminent”.

Quote (#3 Psychiatrist, >20 y PAT exp.): “Good motivation, free will, still some time remaining for integration”.

### 3.4. Importance of Mystical-Type Experience for PC Patients

Mystical-type experiences were considered to be of high value and help relieve death anxiety (9/13, 69%). Two out of the thirteen participants (15%) highlighted a possible similarity between a mystical experience and the process of dying. However, one PAT expert suggested that a mystical-type experience is not required for effective therapy.

Quote (#2 Psychologist, >20 y PAT exp.): “It awakens one to a larger world and often an intuitive sense that ultimately all is well”.

Quote (#10 Psychologist, 0–5 y PAT exp.): “A mystical experience always is a great gift that helps and supports the psychological process. But it is not a requirement for good and helpful therapy. The most important thing is to be able to integrate whatever happens”.

### 3.5. Impact of PAT on PC and Society’s Perception of PC

A majority of the participants (8/13, 62%) hoped that PATPC would change the way society deals with death and dying in general. There is hope that it will be integrated and part of PC as a more holistic approach to end-of-life care, potentially beneficial for relatives and loved ones.

Quote (#11 Psychologist, 0–5 y PAT exp.): “A very important change in paradigm, where we are not just helping people to die by sedating and avoiding the dying experience, which is perceived as only negative, but rather by facing our mortality, to learn to live. Decrease desperation and ignorance about dying”.

Quote (#12 MD, 10–20 y PAT exp.): “It can be and surely must be integrated into late life experiences and can add a much-needed, very broad and deep dimension to it. The extra dimension of love and caring can open heart paths not only for the patient but also their family and loved ones as well”.

(B)Specific Characteristics Distinguishing PAT in PC Patients from PAT in Patients Without Life-Threatening Diseases

### 3.6. Importance of Anxiety, Depression, and Spiritual Distress as Indications for PATPC

Participants considered PATPC also helpful for people without anxiety, depression, and/or spiritual distress (8/13, 62%). Experienced PAT practitioners stressed the personal psychological dimension of psychedelic therapy work, i.e., the possible benefit for the patient if there is a strong need to solve personal or family issues and other psychosocial conflicts. Some also considered the motivation for personal growth and for obtaining a deeper understanding of oneself to be a possible indication for PAT (5/13, 39%).

Quote (#3 Psychiatrist, >20 y PAT exp.): “There are other conditions where PAT could be helpful. Unclear social situation, conflicts with relatives, addictive behavior, isolation and detachment”.

### 3.7. Knowledge of the Concepts of Spiritual Pain (Total Pain) or Existential Distress

Nearly all participants (12/13, 92%) considered it to be important for PAT therapists to be aware of the existence of spiritual and existential distress when accompanying a PC patient in PAT. It was highlighted and considered to be crucial that the therapists have done a sufficient amount of work on themselves and are aware of their own mortality. According to five of the respondents (39%), ideally, the therapists have had their own transcendental experiences to be able to engage deeply with the patient in these existential and transcendental states.

Quote (#8 Psychologist, 2 y PAT exp.): “[Knowledge of the concept of spiritual pain is] important. But even more important is to know how to interact with patients in spiritual pain and being able to be (self) compassionate”.

### 3.8. Repetitive Patterns and Topics in Psychotherapy and/or PATPC

For eight respondents (62%), a common observation was the emergence of family and relational topics during the psychedelic experience (4/13, 31%), as well as the appearance of guilt and regrets about one’s life (4/13, 31%). Five participants (39%) did not reply to this topic.

### 3.9. Difference of Integration Process in PATPC Patients vs. Non-PATPC Patients

With congruity, the limitation of time for integration in the setting of PC was mentioned by 8/13 participants (62%). Associated with that, two of thirteen (15%) noted that people with life-limiting diseases need to focus more on the essential personal topics like biographical and relational questions of life and death, and thus might experience a more intense integration process. Two participants reflected that many palliative patients might have better psychological and emotional resources for integration compared to a primarily psychiatric patient, e.g., have a more stable personality structure, which might have a beneficial impact on integrating even challenging experiences.

Quote (#12 MD, 10–20 y PAT exp.): “It’s maybe more focused on ‘letting go’, just simply as accepting what has been, which is maybe less intellectually time consuming than figuring out how one’s life will have to be adjusted/changed in future years. So, shorter probably”.

(C)Relevant Differences During the Three Phases of PAT (Preparation, Substance Session, Integration)

### 3.10. Preparation

Considering the relatively short amount of time available for PC patients compared to non-PC patients due to life-limiting illness, as reported by six of thirteen (46%) respondents, time for preparation could be proportionately reduced. The general psychological stability of PC patients, in contrast to that of the average psychiatric patient, reported by two of thirteen (15%), may facilitate shortened preparation. Identifying non-medical surroundings (15%) may benefit preparation.

Quote (# 5 MD, 5–10 y PAT exp.) “[In PC there is] greater need for non-medical setting”.

Quote (#12 MD, 10–20 y PAT exp.): “[It is] ok, if it’s in usual setting as moving the subject can cause extra strain/stress/fatigue”.

### 3.11. Substance Session

Working with a medium dose (e.g., corresponding to 100 mcg LSD base) was mentioned in 8/13 replies (62%). The PAT experts preferred either starting low (equivalent to 50–75 mcg LSD base) and going to a medium dose or starting with a medium dose and increasing to a high dose (corresponding to ≥200 mcg LSD base) to enable transpersonal experiences. Generally, starting with a low dose was mentioned by three respondents (23%), consisting of one expert and two non-experts.

A decisive factor in determining the dosage, the intention on topics, and the mental capacity to process the psychedelic experience was named by 12/13. Respondents considered low and medium doses to work best for worldly-oriented intentions (e.g., working on relational issues). If the intention was to precipitate transcendental experiences, a high dose was considered the best choice.

Quote (#8 Psychologist, 2 y PAT exp.): “[It] really depends on person/situation/pre-experiences with psychedelics; medium e.g., if person wants to interact with present relatives/professionals; high to enhance the chance of mystical experience”.

### 3.12. Integration

Brevity is a critical factor for integration in the PC setting and was mentioned by a majority (62%) of participants.

Quote (#1 Psychiatrist, >20 y PAT exp.): “Do you have enough time?”

Quote (#3 Psychiatrist, > 20 y PAT exp.): “The time perspective is different. The patients do not have ‘all the time of the world’”

Addressing more existential themes may also benefit integration, according to 15% of respondents.

Quote (#8 Psychologist, 2 y PAT exp.): “[There is] limited time: faster at the essential questions of life and death”.

The subject of family was brought up by almost half (6/13) of the respondents in different responses to the questionnaire, but the context varied, such that with respect to integration, one suggested that family members be involved.

Quote (#11 Psychologist, 0–5 y PAT exp.): “It may be more helpful for family members and significant others to be involved. Integration may continue for family after death”.

The scope and scale of integration topics could be magnified and include existential themes as they relate to an individual patient’s own mortality (15%).

Quote (#2 Psychologist, >20 y PAT exp.): “[Practitioners should] “focus on what really matters”.

Quote (#8 Psychologist, 2 y PAT exp.): “[Integration ought to be] faster at [getting at] the essential questions of life and death, [which is] sometimes more intense” [but also that] “disappointment [is] difficult to integrate” [despite] “better psychological resources” [of PC patients].

## 4. Discussion

During a workshop on PATPC, we conducted an opt-in questionnaire-based qualitative study, resulting in a low response rate that, however, provided a convenience sample of 13 specialists in the field of PC, oncology, psychiatry/psychology, and psychedelic-assisted psychotherapy on the use of PATPC. There were no differences in overall response between different professions, and the questionnaire gathered information concerning the respondents’ experiences, thoughts, and suggestions for PATPC. Our hypothesis that the unique circumstances in which patients find themselves at the end of life may require an adapted approach in PAT was supported. Experts were asked what special considerations they felt applied to PATPC. With the data collected from our questionnaire, we answer our three initial questions in the order in which they were posed.

### 4.1. Are There Special Considerations on PAT in the PC Setting That Should Be Applied? Yes, Considerations Concerning the Patient’s Health Condition May Have an Impact on Set and Setting

#### 4.1.1. Differences Between the Setting of PATPC vs. Non-PC PAT

This is perhaps relevant in two ways. On the one hand, PATPC patients typically have a shorter life expectancy than non-PATPC patients due to the underlying medical condition, and thus any PAT practitioner will need to consider the time available in planning and applying the therapy. On the other hand, despite these limitations, PATPC clinicians reported PATPC patients to be typically more psychologically stable compared to non-PATPC patients, which may benefit PAT despite the short time horizons.

#### 4.1.2. Parameters Based on Which a Suggestion for PATPC Could Be Considered

A patient’s intrinsic motivation for PAT in association with anxiety and fear of death was reported by about half (54%) of respondents as a crucial factor for the success of PAT. According to participants, additional themes, such as depression, existential distress, and suffering due to unresolved biographical issues, were identified as potential targets for PATPC patients.

#### 4.1.3. Importance of Mystical-Type Experience for Palliative Care Patients

Nearly three-quarters (69%) of participants thought that mystical-type experiences were highly valuable in the relief of death anxiety. Two participants compared the mystical experience to the dying process itself. Indicating some heterogeneity, one participant thought that such an experience is not necessary for effective PAT therapy.

#### 4.1.4. Impact of PAT on PC and Society’s Perception of PC

More than half (62%) of participants expressed their hope that an awareness of PATPC might change how society views death and dying generally. Participants hoped that PAT would eventually become another tool in the PC practitioner’s toolkit, and that PC would in so doing become a more holistic approach to end-of-life care, even to the extent that it would benefit the patient’s family.

### 4.2. Which Specific Characteristics Distinguish PAT in Palliative Patients from PAT in Patients with Mental Illnesses?

#### 4.2.1. Importance of Anxiety, Depression, and Spiritual Distress as Indications for PATPC

Whereas anxiety, depression, and/or spiritual distress are important, most (62%) participants believed that PATPC would also benefit patients without those symptoms. Participants indicated that PATPC emphasized that the personal psychological dimension of psychedelic therapy is present in the resolution of personal or family-related issues and conflicts, or for personal growth.

#### 4.2.2. Knowledge of the Concepts of Spiritual Pain (Total Pain) or Existential Distress

Another characteristic distinguishing PATPC patients from non-PATPC patients considered important by participants is an awareness of spiritual and existential distress that accompanies PC patients. In addition, those who practice PAT may benefit from also having worked on an understanding of their own mortality, including transcendental experiences that would facilitate deep connection to the patient during existential and transcendental states.

#### 4.2.3. Repetitive Patterns and Topics in Psychotherapy and/or PATPC

For about sixty percent of participants, family and relationship topics, including during the psychedelic experience (31%), and guilt and life regrets (31%).

#### 4.2.4. Difference of Integration Process in PATPC Patients vs. Non-PATPC Patients

In the opinion of more than half (62%) of participants, a difference was the shorter timespan for integration with PATPC patients compared to non-PATPC patients. For this reason, the focus might be placed on according to patient perspective; for some relational or symptom issues, for others spiritual/existential experience or awareness. For PATPC patients, this may be made feasible by psychological stability resources available in contrast to psychiatric patients.

### 4.3. To What Extent Are These Differences Relevant During the Three PAT (Preparation, Substance Session, Integration)?

#### 4.3.1. Preparation

Nearly half (46%) of respondents thought that preparation is affected by the life-limited illness that characterizes the PATPC patient population. Identifying the appropriate setting for the PAT session emerged as a consideration in 15% of participants.

#### 4.3.2. Substance Session

There was some appreciable heterogeneity in responses concerning dose, such that medium doses (63%) prevailed generally, whereas starting at a low dose and then escalating to a medium dose, or starting with a medium dose and then increasing to a high dose, was also mentioned. The importance of tailoring dose to the individual and their focus for the dosing session, such that low to medium doses are more suited for worldly-oriented intentions, whereas higher doses are seen as more suitable for the precipitation of transcendental experiences.

#### 4.3.3. Integration

Time available to the practitioner was a factor reported by sixty percent of participants. Addressing more existential themes may also favor successful integration. In addition, the subject of family was addressed by nearly an equal proportion, such that integration might benefit from including family as a theme. Involvement of family members in integration was also suggested.

#### 4.3.4. General Discussion

Anxiety, fear of death, depression, and existential distress were common indications for which experts in this field would consider PAT for palliative patients. Furthermore, experienced PAT-therapists observed that PAT could help people with unsolved personal and relational conflicts sufficiently to relieve their suffering, even in the absence of other common psychiatric symptoms; although, in the medical system, usually the diagnosis of a disorder (e.g., depression) is the prerequisite for indicating a treatment, this seems to an important consideration going forward that is what is the role if any for psychedelic-assisted therapies for people with no psychiatric diagnosis [40]? This question is complicated by the differing regulatory frameworks for PAT by jurisdiction. For example, in Australia, PAT can be prescribed by psychiatrists with a sub-specialty training in PAT, but only for treatment-refractory depression and post-traumatic stress disorder; in contrast, in Switzerland, restricted compassionate use in a medical context is allowed by permit from the federal government. In the EU and UK, psychedelics may only be used for scientific research or limited medical purposes. In Canada, physicians may use psychedelics by permit for certain medical emergencies, whereas in the USA, psychedelics are illegal, but psilocybin and MDMA may be used in the treatment of treatment-resistant depression; various US states have decriminalized possession of psychedelics. How state decriminalization may impact PAT is unclear. We hope this paper will generate hypotheses on safe prescribing in the particular circumstance of PATPC and contribute to an iterative process of recommendation or guideline development, nationally and internationally.

Questionnaire respondents highlighted the intrinsic motivation of the patient as a principal precondition for engaging in PATPC, suggesting that a major indicator for treatment success is one’s own will to undergo an intense, potentially difficult treatment to transform or even transcend one’s own perspectives. Finding a motivation for generating mind-altering procedures and change processes in a situation where time is scarce could be challenging. On the other hand, offering a treatment option to patients who are in a state of existential suffering, desperately seeking change, and willing to actively participate in this change process may be a valuable replenishment for PC specialists.

Most respondents referred to a general structure for PAT with preparatory sessions, dosing session(s), and integrative sessions, as essential in this context. In the preparatory sessions, a key element is to build up a trustworthy therapeutic relationship, to clarify the patient’s expectations and concerns, as well as to clarify the patient’s intention for the psychedelic therapy. Based on this, experts suggested deciding on the dosage of the psychedelic substance. Generally, PAT therapists tend to start with a medium or low dosage if the patient wishes to face personal and relational issues. If the wish is to undergo a transcendental experience, according to the results of our qualitative study, high dosages could be considered. Concerning higher dosages, it is crucial to ensure that the patient is physically and mentally stable enough to hold and process the content of the experience, which sometimes might be overwhelming and challenging. Furthermore, the adverse events of psychedelics are dose-dependent [40,41]. Thus, establishing the appropriate dosage is an essential component for treatment success. Potential short-term (e.g., undesirable processes in the therapeutic relationship, difficult self-experiences) and long-term (e.g., adaptation difficulties, complications in the treatment relationship) adverse outcomes [35], as well as treatment risk-benefit balance, remain, however, largely unexplored and are not well established [36].

In the dosing session, participants of our qualitative study stated that spiritual and transcendental dimensions of a psychedelic experience might appear with a higher prevalence in PC compared to a non-PC population. Thus, therapists working with PATPC might benefit from a profound training and understanding of the spiritual and existential aspects, as well as the concepts of spiritual pain and existential distress. However, participants considered it to be even more important that the therapists have done sufficient inner work on themselves, i.e., on existential questions such as awareness of their own mortality, in order to meet and accompany the patient in an authentic and empathetic manner [42].

After the psychedelic experience, it is crucial to further process the content of the experience, to find meaning, and to integrate it into daily life, i.e., integration of new contents, emotions felt, and maybe even ontological or mystical experiences into pre-existing value systems and worldviews [43,44]. For people with normal life expectancy, this process can take months or even years [45]—indicating that time for adaptation may be limited at the end of life, maybe even limiting the therapeutic potential of PAT in the PC setting.

This paper has several limitations. The sample is from a cohort of clinicians motivated to attend a conference on PATPC, and many are associated with ongoing research in the field. This creates selection bias. The response rate is low, and the sample size is small. Therefore, the respondents, though many of them have a wide scope of experience, may not be representative of the broader field of specialists working with people at the end of life or in psychedelic-assisted therapies. With 13 out of 40 clinicians returning the questionnaire, there is a risk of response bias. In addition, with no external validation or testing of our questionnaire, questions can be leading. The reasons for low response could be related to clinician agreement or disagreement or disinterest regarding the content, but it is not possible to determine. Participants in a workshop, presuming that others are enthusiastic about the topic, may affect each other’s responses. All participants were involved in the same workshop, which may bias responses toward shared attitudes or prior exposure. This is mitigated by the nuanced answers, which seem to suggest a wealth of experience in the topic, which, at face value, supports accepting responses as genuine, although not necessarily representative. In addition, participants included those with zero experience using the PAT. These participants did, however, have experience in the care of patients at the end of life.

The questionnaire’s design might have limited the depth or breadth of information collected, and the questions may not capture the full range of special needs or challenges faced by PC patients in PAT. Since the questions were neither piloted nor validated, there may be issues with their clarity, relevance, or ability to elicit accurate responses. The lack of standardized terminology in the field might have led to varied interpretations of the questions. Incomplete responses or skipped questions can lead to missing data, which might affect the robustness of the conclusions. Nevertheless, to our knowledge, this is the first qualitative study assessing specific considerations for PATPC from PC providers and therapists already experienced with PAT. Future work may include grounded theory methodology, whereby study concepts are developed iteratively as the analysis proceeds, rather than being chosen before the study commences. This would eliminate some bias associated with a questionnaire and add to the foundations of an emerging understanding of the salient issues related to PATPC.

## 5. Conclusions

Our qualitative study among experts in the field of PC, oncology, psychiatry/psychology, including some highly experienced PAT practitioners, highlights relevant specific aspects of PATPC. These include special requirements for training therapists in concepts of existential distress/spiritual pain (total pain). Additionally, PAT experts would consider individually choosing the dosage, depending on the patient’s intention and vulnerability. Sufficient integration of the experience also at the end of life demands special consideration because of the limited lifetime. This study could be a first step towards specific treatment guidelines for PATPC, and, importantly, informs developing practice in determining indications and challenges for PAT in patients towards the end of life. However, a great deal of research on the safety and efficacy of PATPC is needed before such guidelines can be developed. A next step would be to encompass the collection of current practices and experiences in order to develop guidelines, ideally with a mixed methods approach to avoid bias and create a comprehensive foundation for guidelines for practice.

## Figures and Tables

**Table 1 healthcare-13-02275-t001:** Study sample. M.D. = Medical Doctor; PAT = Psychedelic-assisted therapy.

Participant Attributes					
	Psychiatry	Oncology	Palliative Care	M.D. (other specialization)	Psychology
Respondents	2	3	2	2	4
Experience with PAT					
>20 years	2				1
10–20 years				1	
5–10 years		1			
0–5 years		2	2	1	3

## Data Availability

The original contributions presented in this study are included in the article. The data are available in Table A1. Further inquiries can be directed to the corresponding author.

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
