# Peer review of "Psychedelic-Assisted Therapy in Palliative Care—Insights from an International Workshop"

_healthcare, 2025, doi:10.3390/healthcare13182275_

Round 1
Reviewer 1 Report
Comments and Suggestions for Authors
Thank you very much for giving me the opportunity to review this interesting manuscript on such a relevant and topical issue. To improve its clarity and concision, I suggest revising the manuscript with regard to the following aspects:
Introduction:
Some of the references are quite old, particularly in the first paragraphs. Please check if more recent literature is available (e.g. reviews on the wish to hasten death).
METHODS:
Data collection:
What does 'transcribed and sorted according to respondents' level of experience' mean? Wasn't the data already available in written form? How was the data specifically sorted according to the respondents' level of experience, and how did this affect the results and their presentation?
Data analysis:
Thematic analysis: A literature reference would be useful here.
RESULTS
• Study sample: Only 13 out of 40 participants of the face-to-face workshop responded at the survey. This is a quite low response rate and should be mentioned (also in the abstract) and discussed in the discussion section.
• The survey was designed to elicit the participant’s perspectives on the following questions:
A) Are there special considerations on PAT in the PC setting that should be applied?
B) Which specific characteristics distinguish PAT in palliative patients from PAT in patients with mental illnesses without a life-threatening disease?
C) To what extent are these differences relevant during the three phases of PAT (preparation, substance session, integration)?
The Results contains the following structure:
1. General considerations on PATPC
a. Differences between the setting of PATPC vs. Non-PC PAT
b. Parameters based on which a recommendation for PATPC could be considered
c. Importance of anxiety, depression, and spiritual distress as indication for PATP
d. Knowledge of the concepts of spiritual pain (total pain) or existential distress
2. Specific considerations on PATPC
a. Repetitive patterns and topics in Psychotherapy and/or PATPC
b. Difference of integration process in PAT patients with life-limiting disease vs. regular life expectancy
c. Importance of mystical-type experience for palliative patients
d. Dosing recommendations for PATPC
e. Impact of PAT on PC and society's perception of PC
The results section is therewith structured differently to the questions mentioned in the background section and does also not refer to the three phases of PAT mentioned in question C. This makes it difficult to follow the manuscript's thread and should be adjusted somehow.
• Are spiritual distress and spiritual pain the same? (line 147/156) Then I would use the same wording for it.
• Neither the level of experience nor the discipline of the respondents is mentioned when quoting them. This or at least some information should be added to the quotes.
DISCUSSION
• Please add some considerations on the following questions:
o What further research needs arise from the findings?
o What next steps are necessary to develop guidelines on the topic or to move closer to doing so?
Reviewer 2 Report
Comments and Suggestions for Authors
This paper presents a thematic analysis of a questionnaire survey on Psychedelic-Assisted Therapy (PAT) in Palliative Care (PC) to the experts who participated in the international workshop.
This paper clarifies special considerations for PAT in PC.
This paper contains essential findings, although they require a high degree of ethical consideration.
This paper could be improved based on the following 24 comments.
Major comments are 1, 9, 11, 20, and 23.
Minor comments are those other than the above.
1. In the abstract, line 21 says “general recommendations”. Given the content of the text of this paper, it should be written as “expert recommendation” instead of “general recommendation”. Please consider revising the wording.
2. In the abstract, line 25, the term “qualitative analysis” is written. Given the content of the text of this paper, it should be written as “thematic analysis” instead of “qualitative analysis”.
3. In the abstract, in lines 30-31, the term “end of life” is written. Since this wording has no international uniformity, please provide an operative definition where it best fits in your paper.
4. Please separate the results section of your abstract into A), B), and C). This makes it easier to understand the relationship between A), B), and C) described in the methods section.
5. In the abstract, on lines 33-34, it is difficult to understand the meaning of “ importance of sufficient integration”. Please be more specific.
6. In the abstract, line 35, it is difficult to understand the meaning of “transcendental states”. Please be more specific.
7. In the Introduction section, lines 59-78, previous papers have been reviewed. Please include the keywords you used in your review, the search engine's name, such as PubMed, and the study design you employed (whether or not you included expert opinions or case reports).
8. In the Introduction section, lines 59-78, you have included a positive article on the PAT. Please include a negative article on the PAT.
9. In the Methods section, where appropriate, please describe the ethical considerations. Please include the Ethics Committee approval number and whether or not written informed consent was obtained.
10. In the Methods section, on lines 94-98, please state the name of the participant's country, e.g., Europe, the United States, and Australia. Please also indicate the status of PAT approval in that country.
11. In the Methods section, lines 96-98, it states, "Due to the high number of applications for the course, participants were selected based on perceived expertise and motivation." Please list the specific criteria by which you selected the participants.
12. In the Methods section, lines 100-106, please address the following two points: 1) how to reconcile differing opinions among authors, and 2) whether feedback is provided to participants.
13, In the Methods section, lines 111-114, please describe whether or not the analyzed data reached theoretical saturation.
14. The results section, lines 118-119, states that "nine had a medical degree (two psychiatrists, three oncologists, two PC specialists, two not applicable), and four were trained in psychology". Please ensure that the order in which the information is listed is the same as in Table 1.
15. In the Results section, Table 1 would be better titled “Participant Attributes” rather than “Study sample”.
16. In the Results section, after Table 1, between lines 125 and 126, please summarize the results corresponding to the three listed for the purpose of this study, A), B), and C). Example: result A..., result B..., result C....
17. In the Results section, line 127 says “PATPC vs. Non-PC PAT”. Lines 137-138 say "PAT in Palliative Care and non-Palliative Care PAT”. On the other hand, lines 28-29 of the abstract say, "PAT in palliative patients from PAT in patients with mental illnesses”. If these three statements are the same, please make sure that they are written in the same way.
18. In the Discussion section, in the first paragraph, please provide a summary corresponding to the three objectives listed for this study: A), B), and C).
19. Please also add the subsection names in the Discussion section, corresponding to the subsection names listed in the Results section.
The subsection names are as follows:
・General considerations on PATPC
General considerations on PATPC
・Differences between the setting of PATPC vs. Non-PC PAT
・Parameters based on which a recommendation for PATPC could be considered
・(other subsection names should be addressed in the same manner).
20. In the discussion section, on lines 252-254, please remove the word “recommended” from the sentence “If the wish is to undergo a transcendental experience, according to the results of our survey, high dosages would be recommended.” Your study design does not allow anyone to recommend anything. Please state that it is the opinion of an expert.
Please look over this entire paper again, and instead of using the word “recommended” so easily, please state it as an expert opinion.
21. The discussion section, lines 270-271, states, “For people with normal life expectancy, this process can take months or even years.” Please state the basis for this, citing the article.
22. In the Discussion section, line 274 and following, the study's limitations are described. In that section, please include the following. “Participants included those with zero experience using the PAT.”
23. The Conclusion section, lines 294-296, states, "This survey could be a first step towards specific treatment guidelines for PATPC, and, importantly, informs developing practice in determining indications and challenges for PAT in patients toward the end of life. After this sentence, please add the following sentence. "However, a great deal of research on the safety and efficacy of PATPC is needed before such guidelines can be developed."
24. Please prepare an appendix table of the results of this thematic analysis, including not only the Major Domain, but also the Sub-domain and even the lower, Detailed Domain.
Please revise based on my comments or express your disagreement with my comments. Disagreement is welcome, but please do not ignore my comments. Also, please make the text reddish or the background of the text yellow so that it is easy to see where the revisions have been made.
Reviewer 3 Report
Comments and Suggestions for Authors
- The current method lacks critical detail regarding the questionnaire development process.
- No mention of piloting the questionnaire or pre-testing for clarity and bias.
- Include a clear description of the thematic analysis process, with reference to established methodology. who conducted the coding? Were coders independent? How were disagreements resolved? Was software used?
- Was consent obtained? Was the study approved by an ethics committee or declared exempt?
- All participants were involved in the same workshop, which may bias responses toward shared attitudes or prior exposure.
- Was data collection sufficient to reach thematic saturation?
- Page 3: The manuscript states that “comparing the answers of experienced vs. non-experienced participants and therapists with somatic background vs. psychological/psychiatric background showed no relevant differences” (lines 122–123). How these comparisons were conducted?
- The current title, "Table 1 Study sample," is too brief to inform the reader about the content and context of the table. I recommend expanding the title to clearly describe what is being presented — for example, “Table 1. Characteristics of the study sample by professional background and experience with PATPC.”
- Additionally, please ensure that all abbreviations used in the table are explained in a footnote, not in the title.
- Page 5: The theme "Dosing recommendations for PATPC" was not supported by a quote.
- Why did the researchers not ask about patients’ acceptance or refusal of PATPC?
- I recommend including a table that summarizes the main domains, associated themes, and representative participant quotes.
- The study says surveys were given to 40 people, but only 13 responded. That’s about a 32% response rate, which feels pretty low. It would be helpful if the authors could talk about how this might affect their results.
- I recommend describing the study as qualitative rather than strictly survey-based.
Round 2
Reviewer 2 Report
Comments and Suggestions for Authors
The paper was revised based on most of my comments.
The paper was not revised in response to some of my comments.
However, appropriate counterarguments were presented.
As a result, the paper was improved. I want to express my respect for the authors' efforts.
However, I would like to add a comment on the following point.
- Regarding my ninth major comment
The author states the following in lines 128-130 of the revised paper.
“Ethical Considerations. Written consent of all the participants was obtained prior to participation and the questionnaire was anonymized according to Swiss law (HFV Art 25389).”
However, what I am requesting is not the Swiss law number, but the approval number granted by the ethics committee of your affiliated research institution.
Please address my concern.
Reviewer 3 Report
Comments and Suggestions for Authors
hank you for addressing all the comments I raised
